# Risk factors for post-induction hypotension in patients with end-stage renal disease undergoing general anesthesia

Supachai Trikisayaveach[1], Laortip Rattanapittayaporn[2]*, Jatuporn Pakpirom[2], Chutida Sungworawongpana[2], Suttasinee Petsakul[2]

**1** Department of Anesthesiology, Somdech Phra Pinklao Hospital, Bangkok, Thailand, **2** Department of Anesthesiology, Prince of Songkla University, Hatyai, Songkhla, Thailand

* laortip.r@gmail.com

## Abstract

### Background

Patients with end-stage renal disease (ESRD) are at highly susceptible to hemodynamic instability during anesthesia induction. However, data specific to this population remain limited.

### Methods

This retrospective cohort study included 635 adult patients with ESRD who underwent non-cardiac surgery under general anesthesia between 2012 and 2022. Post-induction hypotension was defined as a ≥ 30% reduction in mean arterial pressure (MAP) from baseline within 20 minutes after induction, requiring vasopressor support. Multivariable logistic regression was performed to identify independent predictors.

### Results

The incidence of post-induction hypotension was 44.1% (95% confidence interval (CI), 40.2–47.9). Independent predictors included age > 65 years, chronic oral antidiabetic therapy, higher propofol dose, rapid sequence induction, and emergency surgery. The incidence of hypotension peaked at 20 min after induction. Dialysis-related variables were not significantly associated with hypotension.

### Conclusion

Post-induction hypotension is common in patients with ESRD and associated with advanced age, chronic antidiabetic therapy, higher propofol dose, rapid sequence induction, and emergency surgery.

**Data availability statement:** All relevant data are within the manuscript and its Supporting information files.

**Funding:** The author(s) received no specific funding for this work.

**Competing interests:** No competing interests exist.

## Introduction

The number of patients with end-stage renal disease (ESRD) undergoing surgical procedures has increased substantially in recent years, largely due to improved ESRD survival outcomes and the high prevalence of comorbid conditions, particularly cardiovascular disease. As a result, perioperative management of this population remains a significant clinical challenge.

Post-induction hypotension is a common and clinically important complication associated with adverse outcomes such as myocardial injury, acute kidney injury, stroke, and increased mortality risk. Recent evidence has reinforced the strong association between intraoperative hypotension and postoperative complications, highlighting its clinical significance [1–5].

Recent advances in perioperative monitoring have focused on the early detection and prediction of hypotension, including the development of machine learning–based indices and goal-directed hemodynamic strategies [6,7]. Although hypotension risk factors have been extensively studied in the general surgical population, these findings may not be directly applicable to patients with ESRD as they exhibit distinct physiological alterations, including autonomic dysfunction, vascular stiffness, impaired baroreceptor responses, and altered pharmacokinetics and pharmacodynamics of anesthetic agents. These factors may substantially influence their hemodynamic response to anesthesia induction.

Evidence specifically addressing post-induction hypotension in patients with ESRD remains limited. In particular, the contribution of dialysis-related variables and anesthetic techniques has not been clearly defined. Therefore, this study aimed to determine the incidence and identify risk factors for post-induction hypotension in patients with ESRD undergoing non-cardiac surgery. We hypothesized that ESRD-specific factors contribute to hypotension beyond established predictors observed in the general population.

## Materials and methods

### Study design

This retrospective cohort study was conducted to identify risk factors for post-induction hypotension in patients with ESRD undergoing non-cardiac surgery under general anesthesia at Songklanagarind Hospital. The secondary objective was to determine the incidence of post-induction hypotension in this population.

### Study population

This study included adult patients (≥18 years) with ESRD (glomerular filtration rate <15 mL/min/1.73 m²) receiving chronic hemodialysis who underwent non-cardiac surgery with endotracheal intubation under general anesthesia between 2012 and 2022.

The study protocol was approved by the Human Research Ethics Committee, Faculty of Medicine, Prince of Songkla University (REC 66-290-8-1). Data were accessed on December 8, 2023. The requirement for informed consent was waived due to the retrospective nature of the study.

Patients were excluded if they required more than one attempt at endotracheal intubation, underwent double-lumen endobronchial intubation, were intubated prior to induction, received vasopressor or inotropic support before induction, or had incomplete hemodynamic data. Incomplete data were defined as missing blood pressure or heart rate measurements at baseline or at predefined time points (immediately, and at 5, 10, 15, and 20 min after induction).

## Data collection and variable definitions

Patient data were extracted from the hospital information system by the investigators and a trained anesthetic nurse using standardized data collection forms.

### Independent variables

Collected variables included demographic characteristics (age, sex, body weight, height, body mass index), clinical parameters (dry weight, American Society of Anesthesiologists physical status(ASA)), comorbidities (e.g., hypertension, diabetes mellitus, ischemic heart disease, arrhythmias, structural heart disease, cerebrovascular disease, peripheral vascular disease, thyroid disorders, malignancy, chronic obstructive pulmonary disease, and asthma), and long-term medications.

Preoperative laboratory data included hematocrit, platelet count, blood urea nitrogen, creatinine, electrolytes, bicarbonate, and calcium levels.

Dialysis-related variables included the interval between the last dialysis session and surgery. Baseline hemodynamic parameters were defined as the average of the highest and lowest blood pressure and heart rate recorded preoperatively.

Surgical variables included urgency (elective or emergency) and type of surgery.

Intraoperative variables included induction agents (propofol, etomidate, midazolam, ketamine, thiopental), opioid use, rapid sequence induction, use of regional anesthesia, inhalational agents, and serial hemodynamic measurements at predefined time points (T0, Ti, T5, T10, T15, and T20) and type and total dose of vasopressor (norepinephrine in (µg), ephedrine in (mg), and adrenaline in (µg))

### Dependent variables (Outcomes)

The primary outcome was post-induction hypotension, which was defined as

1. A ≥ 30% decrease in MAP from baseline, and

2. Administration of vasopressor therapy within 20 min after induction.

This definition was based on previous literature demonstrating its association with adverse outcomes.

### Sample size

The sample size was calculated from a formula according to a cohort study using the proportion of outcomes of interest in the exposed group and the proportion of outcomes in the unexposed group. No previous study has investigated the risk factors for post-induction hypotension in patients with ESRD undergoing general anesthesia. The sample size was calculated using data from a previous study that analyzed risk factors for post-induction in patients undergoing general anesthesia (6). The formula is as follows: Five percent significance and 80% were used.

$$n_{exposure} = \left( \frac{z_{1-\frac{\alpha}{2}} \sqrt{\bar{p}\bar{q}\left(1+\frac{1}{r}\right)} + z_{1-\beta} \sqrt{p_1 q_1 + \frac{p_2 q_2}{r}}}{\Delta} \right)^2$$

$$p_1 = P\left(\frac{outcome}{exposure}\right), \quad q_1 = 1 - p_1$$

$$p_2 = P \left( \frac{outcome}{unexposure} \right), \ q_2 = 1 - p_2$$

$$p_1 = p_2 RR, \ \bar{p} = \frac{p_1 + p_2 r}{1 + r}, \ \bar{q} = 1 - \bar{p}$$

$$r = \frac{n_{unexposure}}{n_{exposure}}, \ \Delta = p_1 - p_2$$

**Sample size for the primary outcome: Post-induction hypotension (analgesia with an increased dose of fentanyl)**

Proportion of outcome occurring in exposure group (p1) = 0.103

Proportion of outcome occurring in non-exposure group (p2) = 0.042

Ratio (r) = 1.50

Sample size for exposure group = 228 and non-exposure group = 342

Total sample size = 570, included 10% dropout = 627

**Sample size for primary outcome: Factor of post-induction hypotension (aged 50 years or above)**

Proportion of outcome occurring in the exposure group (p1) = 0.108

Proportion of outcome occurring in the non-exposure group (p2) = 0.02

Ratio (r) = 2.00

Sample size for the exposure group = 82 and non-exposure group = 164

Total sample size = 246, included 10% dropout = 271

**Sample size for the primary outcome: Factors of post-induction hypotension (use of propofol for induction)**

Proportion of outcome occurring in the exposure group (p1) = 0.118

Proportion of outcome occurring in the non-exposure group (p2) = 0.03

Ratio (r) = 3.18

Sample size for the exposure group = 79 and non-exposure group = 251

Total sample size = 330, included 10% dropout = 363

**Sample size for the primary outcome: Pre-induction hypotension factors (baseline MAP<70)**

Proportion of outcome occurring in the exposure group (p1) = 0.211

Proportion of outcome occurring in the non-exposure group (p2) = 0.115

Ratio (r) = 2.63

Sample size for the exposure group = 152 and non-exposure group = 399

Total sample size = 551, included 10% dropout = 606

Therefore, the largest required sample size for the primary outcome was obtained for the "analgesia with increased dose fentanyl." Factor, at a total sample size of 571. The 10% dropout rate was 629.

### Missing data

Multiple imputation was applied for variables with ≤5% missing data. Variables with more than 5% missing data were excluded from the analysis.

### Statistical analysis

Categorical variables are presented as frequencies and percentages, while continuous variables are presented as mean (standard deviation) or median (interquartile range), as appropriate. Univariate analyses were performed using the chi-square test or Fisher's exact test for categorical variables, and Student's t-test or the Wilcoxon rank sum test for continuous variables. Variables with $p < 0.2$ or clinical relevance were included in multivariable logistic regression analysis. Backward stepwise selection based on the Akaike Information Criterion (AIC) was used to derive the final model. Results are reported as odds ratios (ORs) with 95% confidence intervals (CIs). Statistical significance was defined as $p < 0.05$. Model discrimination was assessed using receiver operating characteristic (ROC) curve analysis, and the area under the curve (AUC) was used to evaluate predictive performance.

### ROC curve analysis

Each group was analyzed using multivariate logistic regression. The model was tested using ROC curve analysis to plot the true-positive fraction (sensitivity) versus the false-negative fraction (1-specificity) across various cutoffs to generate an ROC curve. For the summarized entries, the effect of sensitivity and specificity AUC was calculated for the predicted discriminative ability between the post-induction hypotension and the post-induction normotension groups in the logistic regression model. As a result, excellent discrimination was defined by $AUC > 0.9$, good discrimination was defined by AUC 0.75–0.9, moderate or acceptable discrimination was defined by $AUC > 0.6$, and poor or random effect discrimination was defined by $AUC < 0.6$. The model was modified by challenging each factor to determine the optimal AUC.

## Results

A total of 714 patients with ESRD undergoing general anesthesia were identified from the hospital database. After applying the inclusion and exclusion criteria, 635 patients were included in the final analysis (Fig 1). The overall incidence of post-induction hypotension was 44.1% (95% CI, 40.2–47.9).

### Baseline characteristics

Baseline characteristics stratified by hypotension status are presented in Table 1. Patients who developed post-induction hypotension were significantly older, with a higher proportion aged >65 years (P<0.001). Diabetes mellitus was more prevalent in the hypotension group (42.9% vs. 28.2%, P<0.001), whereas the rates of other comorbidities were similar between groups.

Patients in the hypotension group were more likely to receive insulin and oral antidiabetic medication and less likely to receive calcium channel blockers and alpha blockers as long-term therapy than those in the normotension group. Preoperative use of central alpha-2 agonists was associated with a lower incidence of hypotension.

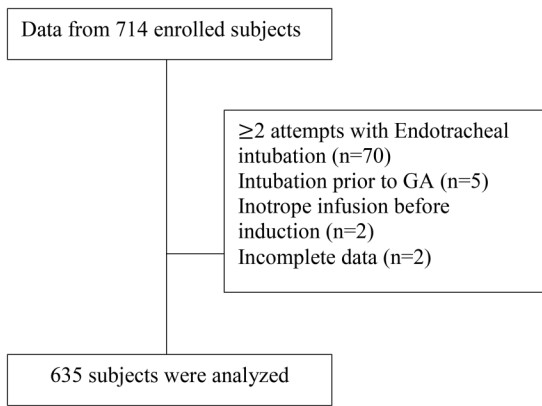

**Fig 1. Flow chart of patient enrollment.**

Preoperative laboratory values showed slightly lower bicarbonate and calcium levels in the hypotension group; however, these differences were not clinically significant.

### Intraoperative factors

Intraoperative characteristics are also summarized in Table 1. Patients who developed hypotension received higher doses of propofol for induction (median 2.5 vs. 2.04 mg/kg, P < 0.001). Rapid sequence induction was more frequently performed in the hypotension group (13.6% vs. 5.4%, P < 0.001). The use of volatile anesthetic agents differed between groups, with a higher proportion of desflurane use in the hypotension group. No significant differences were observed in fentanyl use, fentanyl dose, or nitrous oxide administration.

### Time course of hypotension

The incidence of hypotension increased progressively over time following induction, reaching a peak at 20 min (37.0%). The distribution of hypotensive events across time points according to dialysis interval is shown in Table 2. Patients in the hypotension group exhibited a more pronounced and sustained reduction in MAP compared with those without hypotension (Fig 2). There were no significant differences in MAP trends across different dialysis intervals (Fig 3). Higher doses of propofol and fentanyl were associated with a greater decline in MAP over time (Fig 4).

### Multivariable analysis

Multivariable logistic regression identified the following independent predictors of post-induction hypotension: age > 65 years, chronic oral antidiabetic use, higher propofol dose, rapid sequence induction and emergency surgery. (Table 3)

The predictive model demonstrated acceptable discrimination, with an AUC of 0.732 (95% CI, 0.692–0.771) (Fig 5).

## Discussion

### Principal findings

In this study, we found that post-induction hypotension occurred in 44.1% of patients with ESRD, indicating a high burden of hemodynamic instability in this population. Several independent predictors were identified, including

**Table 1. Demographic data and patient characteristics of the post-induction hypotension and normotension groups.**

| Characteristic | Normotension (n = 355)[a] | Hypotension (n = 280)[a] | p-value[b] |
|---|---|---|---|
| Sex | | | 0.316 |
| Male | 184(51.8%) | 133 (47.5%) | |
| Female | 171(48.2%) | 147(52.5%) | |
| Age group | | | <0.001 |
| 18–64 years | 286 (80.6%) | 163 (58.2%) | |
| 65–74 years | 45 (12.7%) | 68 (24.3%) | |
| > 74 years | 24 (6.8%) | 49 (17.5%) | |
| ASA classification | | | 0.349 |
| 3 | 351 (98.9%) | 274 (97.9%) | |
| 4 | 4 (1.1%) | 6 (2.1%) | |
| Bodyweight (Kg)** | 57 (50, 65) | 57 (50, 65) | 0.435 |
| BMI (kg/m2) ** | 21.9 (19.9,24.9) | 22.65 (20.3, 25.5) | 0.271 |
| Volume status | | | |
| Hypovolemia (Delta body weight <0) | 10(2.82) | 7(2.5) | 1 |
| Hypervolemia (Delta body weight >0) | 345(97.18) | 273(97.5) | |
| Days of dialysis before operation (days) | 2(1,2) | 2(1,2) | 0.199 |
| Underlying disease | | | |
| Hypertension | 327 (92.1%) | 263 (93.3%) | 0.466 |
| Diabetes mellitus | 100 (28.2%) | 120 (42.9%) | <0.001 |
| Ischemic heart disease | 39 (11%) | 38 (13.57%) | 0.385 |
| Significant arrhythmia | 26 (7.3%) | 20 (7.1%) | 1 |
| Structural heart disease | 15 (4.2%) | 19 (6.79%) | 0.213 |
| Cerebrovascular disease | 19 (5.35%) | 24 (8.57%) | 0.149 |
| Peripheral vascular disease | 7 (1.97%) | 8 (2.86%) | 0.641 |
| Hyperthyroidism | 3 (0.85%) | 1 (0.36%) | 0.634 |
| Hypothyroidism | 6 (1.69%) | 6 (2.14%) | 0.903 |
| Cancer | 6 (1.69%) | 5 (1.79%) | 1 |
| Chronic Obstructive Pulmonary Disease | 4 (1.13%) | 1 (0.36%) | 0.390 |
| Asthma | 3 (0.85%) | 5 (1.79%) | 0.311 |
| Long-term medication | | | |
| Beta-blockers | 166 (46.76%) | 143 (51.07%) | 0.318 |
| Calcium channel blockers | 223 (62.82%) | 149 (53.21%) | 0.018 |
| Angiotensin Converting Enzyme Inhibitors (ACEIs) | 9 (2.54%) | 7 (2.5%) | 1 |
| Angiotensin II Receptor Blockers (ARB) | 76 (21.41%) | 62 (22.14%) | 0.900 |
| Diuretics | 84 (23.66%) | 81 (28.93%) | 0.158 |
| Direct vasodilators | 144 (40.56%) | 116 (41.43%) | 0.890 |
| Alpha blockers | 106 (29.86%) | 58 (20.71%) | 0.011 |
| Central alpha 2 agonists | 14 (3.94%) | 4 (1.43%) | 0.098 |
| Insulin | 27 (7.61%) | 42 (15%) | 0.005 |
| Oral antidiabetics | 10 (2.82%) | 26 (9.29%) | < 0.001 |
| Thyroid therapy | 5 (1.41%) | 4 (1.43%) | 1 |
| Preoperative-medication | | | |
| Beta-blockers | 121 (34.08%) | 108 (38.57%) | 0.278 |
| Calcium channel blockers | 166 (46.76%) | 118 (42.14%) | 0.279 |
| ACEIs | 2 (0.56%) | 3 (1.07%) | 0.659 |

*(Continued)*

**Table 1.** (Continued)

| Characteristic | Normotension (n = 355)[a] | Hypotension (n = 280)[a] | p-value[b] |
|---|---|---|---|
| ARB | 21 (5.92%) | 19 (6.79%) | 0.777 |
| Diuretics | 24 (6.76%) | 29 (10.36%) | 0.138 |
| Direct vasodilators | 91 (25.63%) | 72 (25.71%) | 1 |
| Alpha blockers | 22 (6.2%) | 17 (6.07%) | 1 |
| Central alpha 2 agonists | 10 (2.82%) | 1 (0.36%) | 0.028 |
| Preoperative laboratory | | | |
| Hematocrit (Hct) (%)*** | 32.58 (5.81) | 33.38 (6.17) | 0.096 |
| Blood Urea Nitrogen (BUN) (mg/dL)** | 36.5 (24.55, 50.7) | 34.65 (24.85, 46.45) | 0.307 |
| Creatinine (Cr) (mg/dL)** | 7.3 (5.3, 9.8) | 6.8 (5, 9.1) | 0.049 |
| Bicarbonate (mmol/L)** | 25.2 (23, 27.1) | 24.3 (22.1, 26.1) | <0.001 |
| Calcium (mmol/L)** | 10 (9.2, 10.5) | 9.7 (9, 10.3) | 0.033 |
| Intraoperative | | | |
| Propofol usage | 351 (98.87%) | 274 (97.86%) | 0.349 |
| Dose of propofol at induction** (mg/kg) | 2.04 (1.55,2.78) | 2.5 (1.85,3.09) | <0.001 |
| Fentanyl usage | 345 (97.18%) | 271 (96.79%) | 0.954 |
| Dose of fentanyl at induction*** (mcg/kg) | 1.55 (0.6) | 1.52 (0.57) | 0.515 |
| Type of surgery (Emergency) | 53 (14.93%) | 30 (10.71%) | 0.148 |
| Rapid sequence induction technique | 19 (5.35%) | 38 (13.57%) | <0.001 |
| Nitrous oxide used | 4 (1.13%) | 3 (1.07%) | 1 |
| Sevoflurane | 257 (72.39%) | 164 (58.57%) | <0.001 |
| Desflurane | 98(27.61%) | 116(41.43%) | <0.001 |

[a]n (%), [b]Pearson's Chi-squared test; Wilcoxon rank sum test; Fisher's exact test.

** Median (IQR), *** Mean (SD).

**Table 2. Hypotension events at any time points within 20 min after induction period according to day of dialysis interval.**

| Dialysis interval (days) | Time points | | | | |
|---|---|---|---|---|---|
| | T1 | T5 | T10 | T15 | T20 |
| 1 day | 8 | 21 | 31 | 53 | 60 |
| 2 days | 8 | 18 | 22 | 54 | 57 |
| At least 3 days | 2 | 6 | 4 | 20 | 28 |
| Total | 18 (4.6%) | 45 (11.5%) | 57 (14.5%) | 127 (32.4%) | 145 (37.0%) |

advanced age, chronic antidiabetic therapy, higher propofol dosage, rapid sequence induction, and emergency surgery.

Notably, hypotension demonstrated a delayed peak at approximately 20 min after induction, which differs from patterns typically reported in the general population.

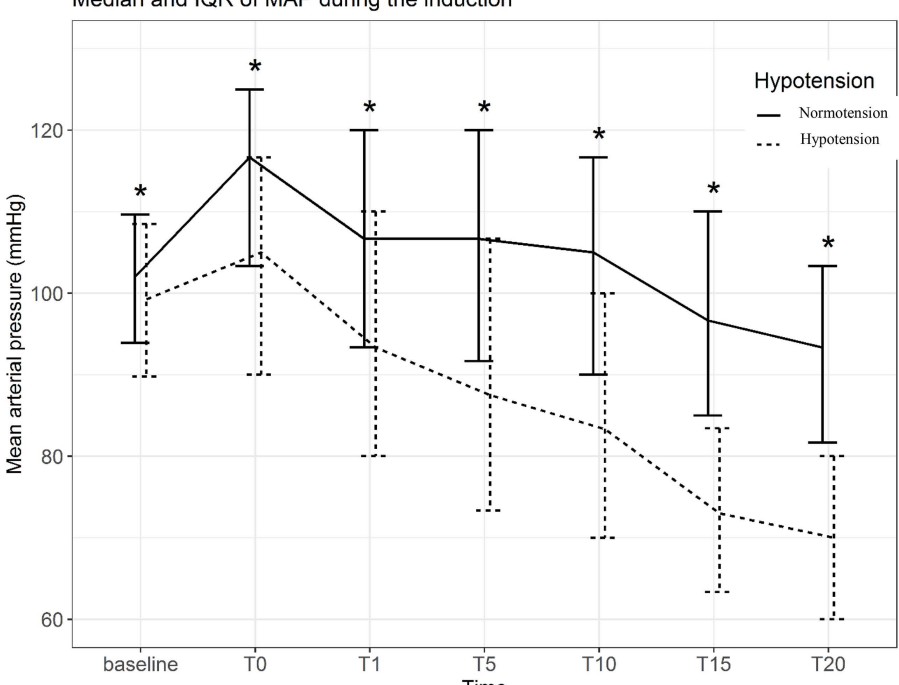

**Fig 2. Median MAP during the induction.**

## Interpretation of findings

The high incidence of hypotension observed in this study likely reflects the combined effects of autonomic dysfunction, reduced vascular compliance, and impaired cardiovascular reserve in patients with ESRD. These physiological changes limit the body's ability to compensate for anesthetic-induced vasodilation and myocardial depression [8].

The delayed onset of hypotension may be explained by altered pharmacokinetics and pharmacodynamics of anesthetic agents in ESRD, particularly propofol. Reduced protein binding, changes in volume of distribution, and impaired drug clearance may contribute to prolonged hemodynamic effects.

Importantly, accumulating evidence suggests that even brief episodes of intraoperative hypotension are associated with adverse outcomes, including acute kidney injury and increased mortality [9,10]. This underscores the clinical importance of identifying intraoperative risks, optimal perioperative strategies, and improved risk analysis for populations such as patients with ESRD.

Advanced age was a strong predictor of hypotension, consistent with previous literature. Age-related reductions in baroreceptor sensitivity and cardiac reserve are likely exacerbated by uremia-associated vascular and myocardial changes in patients with ESRD.

The association between chronic antidiabetic therapy and hypotension likely reflects underlying autonomic neuropathy rather than a direct pharmacological effect. Similarly, rapid sequence induction may exacerbate hypotension due to limited time for hemodynamic compensation.

Anesthetic factors also played a significant role. Higher propofol doses and rapid sequence induction were independently associated with hypotension. Propofol-induced vasodilation and myocardial depression are well established [11], and these effects may be amplified in ESRD due to altered drug handling. Rapid sequence induction limits dose titration and may exacerbate hemodynamic instability, particularly in patients with limited physiological reserve [12].

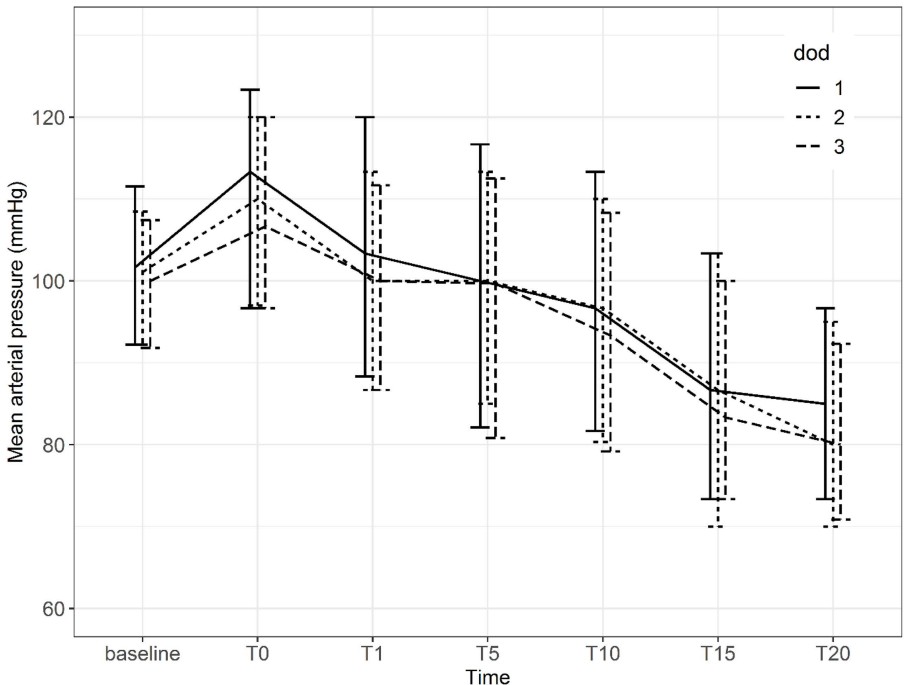

**Fig 3. Median MAP during the induction by the number of pre-op dialysis days.**

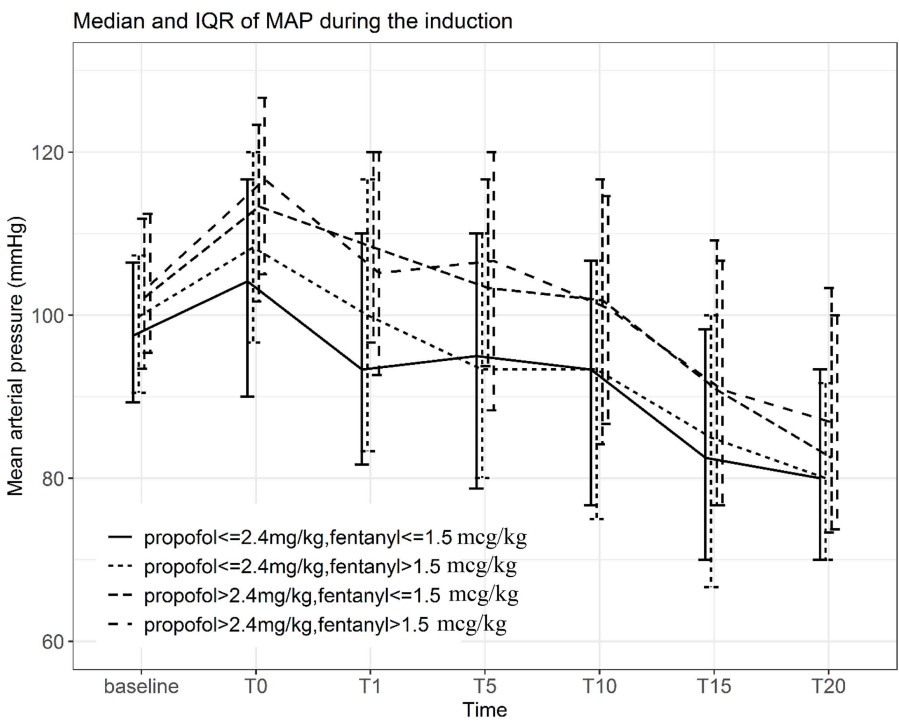

**Fig 4. Median MAP during the induction according to propofol and fentanyl dose.**

**Table 3. Risk factors for hypotension occurring at one or more time points within 20 min after intubation, based on multivariate logistic regression.**

| Factors | Crude OR (95% CI) | Adj. OR (95% CI) | P (Wald's test) | p-value |
|---|---|---|---|---|
| Age group Ref. = 18–64 65–74 75–88 | 2.65 (1.73,4.05) 3.71 (2.18,6.33) | 2.04 (1.29, 3.24) 2.39 (1.31,4.35) | 0.002 0.004 | <0.001 |
| Calcium channel blockers as long-term medication | 1.51 (1.09,2.08) | 1.34 (0.94,1.9) | 0.101 | 0.101 |
| Oral antidiabetics as long-term medication | 3.57 (1.69,7.55) | 2.83 (1.29,6.18) | 0.009 | 0.007 |
| Diuretic as premedication | 8.01 (1.02,62.9) | 5.61 (0.68,46.65) | 0.11 | 0.052 |
| Dose of propofol (mg) | 0.992 (0.9889,0.9952) | 0.9944 (0.9908,0.998) | 0.002 | 0.002 |
| Rapid sequence induction technique | 2.80 (1.55,5.05) | 3.83 (2.01,7.30) | < 0.001 | <0.001 |
| Desflurane | 0.54 (0.38,0.75) | 0.69 (0.48,1) | 0.049 | 0.050 |
| Type of surgery: emergency | 1.49 (0.91,2.44) | 2.40 (1.37,4.19) | 0.003 | 0.002 |
| Volume status | 1.58 (0.53,4.67) | 2.58 (0.79,8.45) | 0.119 | 0.109 |

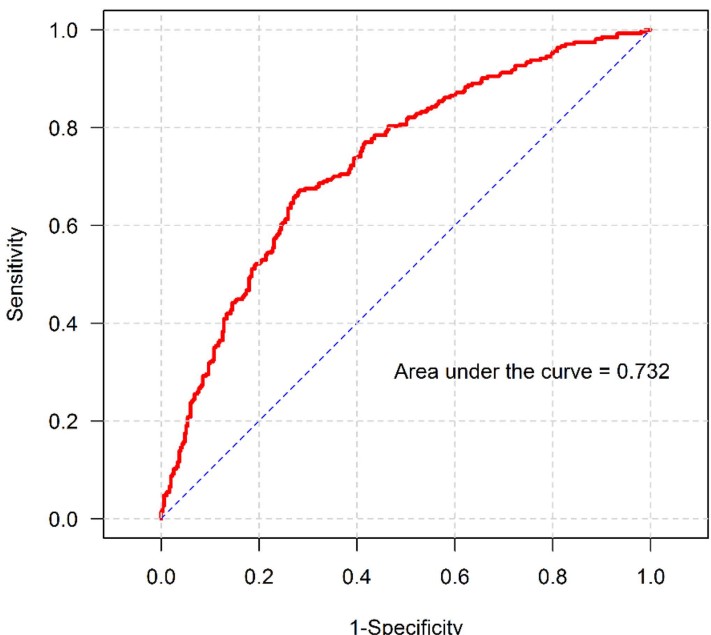

**Fig 5. Area under the curve of the multivariate model.**

## Dialysis-related factors and volume status

Interestingly, dialysis-related variables were not significantly associated with post-induction hypotension. This finding suggests that traditional assumptions regarding volume status based on dialysis timing or modality may not reliably predict hemodynamic responses during anesthesia induction.

## Clinical implications

These findings have important implications for anesthetic management in patients with ESRD. Careful titration of induction agents, avoidance of excessive dosing, and early use of vasopressors should be considered. In addition, extended hemodynamic monitoring beyond the immediate post-induction period may be warranted.

## Strengths and limitations

This study has several strengths, including a relatively large sample size and a focused analysis of a high-risk population. However, several limitations should be acknowledged. The retrospective design may introduce selection bias, and blood pressure measurements were not continuously recorded; and long-term outcomes were not evaluated.

## Conclusions

Post-induction hypotension is highly prevalent in patients with ESRD and is influenced by advanced age, chronic antidiabetic therapy, higher propofol dosage, rapid sequence induction, and emergency surgery. The delayed pattern of hypotension observed in this study highlights the need for prolonged vigilance during the peri-induction period. Future prospective studies are warranted to validate these findings and optimize management strategies.

## Supporting information

**S1 File. This is S1 Table Title. This is the S1 Table legend.**
(XLSX)

## Author contributions

**Data curation:** Supachai Trikisayaveach.

**Methodology:** Supachai Trikisayaveach.

**Supervision:** Jatuporn Pakpirom, Suttasinee Petsakul.

**Writing – original draft:** Laortip Rattanapittayaporn, Supachai Trikisayaveach.

**Writing – review & editing:** Laortip Rattanapittayaporn, Chutida Sungworawongpana.

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
