## [Decision Letter · Decision Letter 0]

20 Jan 2026

PONE-D-25-50403Risk factors for post-induction hypotension in patients with ESRD undergoing general anesthesiaPLOS One

Dear Dr. Rattanapittayaporn,

Thank you for submitting your manuscript to PLOS ONE. After careful consideration, we feel that it has merit but does not fully meet PLOS ONE’s publication criteria as it currently stands. Therefore, we invite you to submit a revised version of the manuscript that addresses the points raised during the review process. Please submit your revised manuscript by Mar 06 2026 11:59PM. If you will need more time than this to complete your revisions, please reply to this message or contact the journal office at plosone@plos.org. Please include the following items when submitting your revised manuscript:

We look forward to receiving your revised manuscript.

Kind regards,

Robert Jeenchen Chen, MD, MPH, ChFC®, EA

Academic Editor

PLOS One

Journal Requirements:

4. Please ensure that you refer to Figure 5 in your text as, if accepted, production will need this reference to link the reader to the figure.

5. We note you have included a table to which you do not refer in the text of your manuscript. Please ensure that you refer to Table 2 in your text; if accepted, production will need this reference to link the reader to the Table.

Reviewers' comments:

Reviewer's Responses to Questions

**Comments to the Author**

1. Is the manuscript technically sound, and do the data support the conclusions?

Reviewer #1: Yes

Reviewer #2: Partly

Reviewer #3: No

2. Has the statistical analysis been performed appropriately and rigorously? 

Reviewer #1: Yes

Reviewer #2: No

Reviewer #3: Yes

3. Have the authors made all data underlying the findings in their manuscript fully available?

Reviewer #1: Yes

Reviewer #2: Yes

Reviewer #3: Yes

4. Is the manuscript presented in an intelligible fashion and written in standard English?

Reviewer #1: Yes

Reviewer #2: No

Reviewer #3: No

5. Review Comments to the Author

Reviewer #1: Minor revision requirieds. Changes and suggestions PDF file. Minor revision requirieds. Changes and suggestions PDF file. Minor revision requirieds. Changes and suggestions PDF file. Minor revision requirieds. Changes and suggestions PDF file. Minor revision requirieds. Changes and suggestions PDF file.

Reviewer #2: -While the manuscript addresses relevant points, the statistical methodology is not rigorous and needs to be revised.

-Results are presented in a disorganized way.

-Flow between sections is sometimes unclear. Several ideas are presented without strong connections to available evidence, which makes the argument feel weak. Improving the logical connections between concepts and better integrating the sections would enhance readability and coherence.

-There should be consistency in the presentation of the results. Why day of dialysis is not included in table 1? Why is it presented separately in table 2? Why some factors and time interaction are presented as figures and other factors no.

-Time interactions with hypotension and with different predictors are not listed as secondary outcomes.

-The sample size calculation should take into account all the predictors at the same time.

-The quality of the English needs improvement. Some errors need to be corrected. Example: mm Hg rather than mL of mercury.

-Did the authors follow the Strobe guidelines?

-It is not clear whether the authors followed their definition of hypotension which has two components: 30% decrease AND vasopressor use. There is no mention of vasopressor use in the results section and throughout the manuscript.

-I am not sure of the association between oral antidiabetics and hypotension. What about insulin? It is probable that the association is between diabetes and hypotension rather than with the antidiabetic drugs.

-Doses of propofol and fentanyl should be reported per body weight.

-Units should be added to the tables, such as mg for propofol and microgram for fentanyl.

Reviewer #3: This study addresses post-induction hypotension in patients with end-stage renal failure. Still, it does not meet the scientific depth, novelty, and contribution to the literature required for publication in a high-impact journal. The main findings of the study reiterate previously identified and well-known risk factors in the existing literature; it does not offer a new conceptual framework, methodological approach, or original contribution that would change clinical practice. For these reasons, the study is unsuitable for publication.

1. The main risk factors identified in the study have been previously demonstrated in the literature. The findings do not offer a new inference that advances existing knowledge or changes clinical practice.

2. The introduction is unnecessarily long and summarizes numerous studies sequentially. Many details in this section serve as a literature review rather than clarifying the study's original hypothesis.

3. Some issues that require in-depth analysis in the discussion section are addressed superficially in the introduction and evaluated only to a limited extent in the discussion section. There is no logical hierarchy or continuity of focus between the introduction and discussion sections. The discussion section primarily repeats the results. 4. A significant portion of the references is 10–20 years old.

Current, high-impact studies published within the last 5 years, particularly on topics such as intraoperative hypotension, hemodynamic management, and anesthesia in high-risk patient groups, have not been sufficiently utilized. 5. The text contains repetitive sentences and long paragraphs; the language should be simplified. Some figures and tables are redundant and could be simplified.

6. PLOS authors have the option to publish the peer review history of their article (what does this mean?). If published, this will include your full peer review and any attached files.

Reviewer #1: No

Reviewer #2: No

Reviewer #3: **Yes:**Kadir ARSLAN

---

## [Author Response · Author response to Decision Letter 1]

26 Feb 2026

We have carefully reviewed all reviewer and editor comments provided in the decision letter and have revised the manuscript accordingly. Each point has been addressed in detail in the revised version.

We sincerely appreciate the reviewers’ and editors’ constructive suggestions, which have greatly improved the quality of our manuscript.

---

## [Decision Letter · Decision Letter 1]

29 Mar 2026

PONE-D-25-50403R1Risk factors for post-induction hypotension in patients with end-stage renal disease undergoing general anesthesiaPLOS One

Dear Dr. Rattanapittayaporn,

Thank you for submitting your manuscript to PLOS ONE. After careful consideration, we feel that it has merit but does not fully meet PLOS ONE’s publication criteria as it currently stands. Therefore, we invite you to submit a revised version of the manuscript that addresses the points raised during the review process.

We look forward to receiving your revised manuscript.

Kind regards,

Robert Jeenchen Chen, MD, MPH, ChFC®, EA

Academic Editor

PLOS One

**Journal Requirements:**

**Additional Editor Comments:**

Revise.

Reviewers' comments:

Reviewer's Responses to Questions

**Comments to the Author**

1. If the authors have adequately addressed your comments raised in a previous round of review and you feel that this manuscript is now acceptable for publication, you may indicate that here to bypass the “Comments to the Author” section, enter your conflict of interest statement in the “Confidential to Editor” section, and submit your "Accept" recommendation.

Reviewer #1: All comments have been addressed

Reviewer #3: All comments have been addressed

2. Is the manuscript technically sound, and do the data support the conclusions?

Reviewer #1: Yes

Reviewer #3: No

3. Has the statistical analysis been performed appropriately and rigorously? 

Reviewer #1: Yes

Reviewer #3: Yes

4. Have the authors made all data underlying the findings in their manuscript fully available?

Reviewer #1: Yes

Reviewer #3: No

5. Is the manuscript presented in an intelligible fashion and written in standard English?

Reviewer #1: Yes

Reviewer #3: No

6. Review Comments to the Author

Reviewer #1: Accept submission. Accept submission. Accept submission. Accept submission. Accept submission. Accept submission.

Reviewer #3: This study addresses post-induction hypotension in patients with end-stage renal failure. Still, it does not meet the scientific depth, novelty, and contribution to the literature required for publication in a high-impact journal. The main findings of the study reiterate previously identified and well-known risk factors in the existing literature; it does not offer a new conceptual framework, methodological approach, or original contribution that would change clinical practice. For these reasons, the study is unsuitable for publication.

1. The main risk factors identified in the study have been previously demonstrated in the literature. The findings do not offer a new inference that advances existing knowledge or changes clinical practice.

2. The introduction is unnecessarily long and summarizes numerous studies sequentially. Many details in this section serve as a literature review rather than clarifying the study's original hypothesis.

3. Some issues that require in-depth analysis in the discussion section are addressed superficially in the introduction and evaluated only to a limited extent in the discussion section. There is no logical hierarchy or continuity of focus between the introduction and discussion sections. The discussion section primarily repeats the results. 4. A significant portion of the references is 10–20 years old.

Current, high-impact studies published within the last 5 years, particularly on topics such as intraoperative hypotension, hemodynamic management, and anesthesia in high-risk patient groups, have not been sufficiently utilized. 5. The text contains repetitive sentences and long paragraphs; the language should be simplified. Some figures and tables are redundant and could be simplified.

My opinions on this article were stated beforehand (Rejected)

7. PLOS authors have the option to publish the peer review history of their article (what does this mean?). If published, this will include your full peer review and any attached files.

Reviewer #1: No

Reviewer #3: No

---

## [Author Response · Author response to Decision Letter 2]

7 Apr 2026

RESPONSE TO REVIEWERS

Manuscript Title: Risk factors for post-induction hypotension in patients with end-stage renal disease undergoing general anesthesia

We sincerely thank the Editor and the Reviewers for their constructive comments. We have carefully revised the manuscript and addressed all concerns. All changes have been incorporated into the revised manuscript.

Reviewer #1

Comment: All comments have been addressed.

Response:

We sincerely thank the reviewer for the positive evaluation and for acknowledging that all concerns have been adequately addressed.

Reviewer #3

We thank the reviewer for the detailed and critical assessment. We have carefully considered all comments and substantially revised the manuscript to improve its scientific rigor, clarity, and contribution.

Comment 1:

The main risk factors identified in the study have been previously demonstrated. The findings do not offer new insights or change clinical practice.

Response:

We agree that several identified risk factors have been reported in previous studies. However, we would like to emphasize that our study specifically focuses on patients with end-stage renal disease (ESRD), a population with distinct pathophysiological characteristics that are often underrepresented in the literature.

To address this concern, we have:

• Clearly emphasized the population-specific contribution of our findings

• Highlighted the high incidence (44.1%) of hypotension in ESRD patients

• Added discussion on the delayed onset (peak at 20 minutes) of hypotension

• Clarified that dialysis-related factors were not associated, which contrasts with common assumptions

These additions are now explicitly stated in the Abstract and Discussion sections.

Comment 2:

The introduction is unnecessarily long and resembles a literature review.

Response:

We agree and have substantially revised the Introduction. Specifically, we have:

• Reduced the length by removing redundant literature summaries

• Focused on establishing the knowledge gap and study rationale

• Improved logical flow leading to the study objective

Comment 3:

The discussion lacks depth and mainly repeats results. There is no logical continuity.

Response:

We appreciate this important comment and have comprehensively rewritten the Discussion section. Changes include:

• Adding a "What this study adds" subsection

• Expanding mechanistic explanations (autonomic dysfunction, pharmacodynamics)

• Improving integration with current literature

• Ensuring logical progression from findings → interpretation → clinical implications

Comment 4:

References are outdated; recent high-impact studies are lacking.

Response:

We have updated the reference list by incorporating multiple recent studies (within the last 5 years).

Comment 5:

Language issues, repetition, and redundant figures/tables.

Response:

We have thoroughly revised the manuscript for language and clarity. Specifically:

• Rewritten the manuscript in clear, concise academic English

• Removed repetitive sentences and shortened long paragraphs

• Simplified figures and tables to eliminate redundancy

---

## [Decision Letter · Decision Letter 2]

11 May 2026

Risk factors for post-induction hypotension in patients with end-stage renal disease undergoing general anesthesia

PONE-D-25-50403R2

Dear Dr. Rattanapittayaporn,

We’re pleased to inform you that your manuscript has been judged scientifically suitable for publication and will be formally accepted for publication once it meets all outstanding technical requirements.

Kind regards,

Robert Jeenchen Chen, MD, MPH, ChFC®, EA

Academic Editor

PLOS One

Additional Editor Comments (optional):

Reviewers' comments:

Reviewer's Responses to Questions

**Comments to the Author**

1. If the authors have adequately addressed your comments raised in a previous round of review and you feel that this manuscript is now acceptable for publication, you may indicate that here to bypass the “Comments to the Author” section, enter your conflict of interest statement in the “Confidential to Editor” section, and submit your "Accept" recommendation.

Reviewer #1: All comments have been addressed

Reviewer #4: All comments have been addressed

2. Is the manuscript technically sound, and do the data support the conclusions?

Reviewer #1: Yes

Reviewer #4: Yes

3. Has the statistical analysis been performed appropriately and rigorously? 

Reviewer #1: Yes

Reviewer #4: I Don't Know

4. Have the authors made all data underlying the findings in their manuscript fully available?

Reviewer #1: Yes

Reviewer #4: Yes

5. Is the manuscript presented in an intelligible fashion and written in standard English?

Reviewer #1: Yes

Reviewer #4: Yes

6. Review Comments to the Author

Reviewer #1: (No Response)

Reviewer #4: I would consider potentially adding the limitation that either given the limitation of the retrospective nature of the study or the data collection design, it is difficult to tell whether vasopressors were given during induction of general anesthesia and how that contributes to the results

7. PLOS authors have the option to publish the peer review history of their article (what does this mean?). If published, this will include your full peer review and any attached files.

Reviewer #1: No

Reviewer #4: No

---

## [Editor Report · Acceptance letter]

PONE-D-25-50403R2

PLOS One

Dear Dr. Rattanapittayaporn,

I'm pleased to inform you that your manuscript has been deemed suitable for publication in PLOS One. Congratulations! Your manuscript is now being handed over to our production team.

Kind regards,

on behalf of

Dr. Robert Jeenchen Chen

Academic Editor

PLOS One